# Reduced Sympathetic Reserve Detectable by Heart Rate Response after Dipyridamole in Anginal Patients with Normal Coronary Arteries

**DOI:** 10.3390/jcm11010052

**Published:** 2021-12-23

**Authors:** Lauro Cortigiani, Clara Carpeggiani, Laura Meola, Ana Djordjevic-Dikic, Francesco Bovenzi, Eugenio Picano

**Affiliations:** 1Cardiology Division, San Luca Hospital, 55100 Lucca, Italy; laurameola91@gmail.com (L.M.); f.bovenzi@tin.it (F.B.); 2CNR Institute of Clinical Physiology, 56125 Pisa, Italy; clara@ifc.cnr.it (C.C.); picano@ifc.cnr.it (E.P.); 3Cardiology Clinic, Medical School, University of Belgrade, 11000 Belgrade, Serbia; skali.ana7@gmail.com

**Keywords:** dipyridamole, heart rate, INOCA, prognosis, stress echocardiography

## Abstract

**Background**. Patients with ischemia and normal coronary arteries (INOCA) may show abnormal cardiac sympathetic function, which could be unmasked as a reduced heart rate reserve (HRR) during dipyridamole stress echocardiography (SE). **Objectives**. To assess whether HRR during dipyridamole SE predicts outcome. **Methods**. Dipyridamole SE was performed in 292 patients with INOCA. HRR was measured as peak/rest heart rate and considered abnormal when ≤1.22 (≤1.17 in presence of permanent atrial fibrillation). All-cause death was the only endpoint. **Results**. HRR during SE was normal in 183 (63%) and abnormal in 109 patients (37%). During a follow-up of 10.4 ± 5.5 years, 89 patients (30%) died. The 15-year mortality rate was 27% in patients with normal and 54% in those with abnormal HRR (*p* < 0.0001). In a multivariable analysis, a blunted HRR during SE was an independent predictor of outcome (hazard ratio 1.86, 95% confidence intervals 1.20–2.88; *p* = 0.006) outperforming inducible ischemia. **Conclusions**. A blunted HRR during dipyridamole SE predicts a worse survival in INOCA patients, independent of inducible ischemia.

## 1. Introduction

Patients with ischemia and normal coronary arteries (INOCA) are highly prevalent in contemporary populations, show a relatively higher risk for cardiac events compared to reference subjects and their management is often unsatisfactory [1]. The heterogeneity of the underlying mechanisms is difficult to recognize in the individual patient, and therapy cannot be tailored to the prevailing mechanism [2]. One of the proposed mechanisms is autonomic dysfunction [3], which may also interact with the other established pathophysiological pathway of coronary microvascular dysfunction [4]. Cardiac sympathetic reserve can be assessed through the simple evaluation of the ratio of peak to rest heart rate, named heart rate reserve (HRR), during a dipyridamole test, which is frequently used in INOCA to assess coronary microcirculatory function [5]. Dipyridamole stress may serve to unmask ischemia as inducible regional wall motion abnormalities (RWMA) and also as a noninvasive probe of cardiac autonomic dysfunction [6,7]. Endogenous adenosine accumulates with dipyridamole and stimulates excitatory adenosine A2a receptors present on cardiac afferent neurons with an increase in sympathetic outflow [8,9,10]. We hypothesized that a reduced increase in heart rate during dipyridamole SE identifies the cardiac autonomic unbalance phenotype in INOCA patients and predicts an unfavorable outcome, in a manner independent from inducible ischemia.

## 2. Materials and Methods

**Patient population.** From 1998 to 2019, we prospectively acquired and retrospectively analyzed 292 patients (age 64 ± 11 years, ejection fraction 57 ± 7%) with INOCA, referred to dipyridamole SE for diagnostic purposes. Inclusion criteria for INOCA diagnosis were history of stable typical or atypical chest pain over months or years, ECG or perfusion imaging evidence of ischemia during stress, and angiographically normal coronary arteries (<50% diameter stenosis in any coronary vessel). 

Exclusion criteria at entry were inadequate acoustic window, angiographically significant coronary artery disease, previous myocardial revascularization, previous myocardial infarction, baseline global left ventricular dysfunction (ejection fraction < 40%), or other prognosis limiting comorbidities, as previously described in detail [6]. All readers followed recommendations of major echocardiography scientific societies in image acquisition and analysis [11,12], and were accredited via web with high interobserver concordance in assessment of RWMA (>90%) and HRR (>99%), as previously described [13]. 

**Stress protocol.** Dipyridamole was infused at the dose of 0.84 mg/kg for 6 or 10 min, according to a protocol previously described in detail [13]. Ischemia was a new or worsening RWMA in at least 2 contiguous segments [13]. HRR (peak/rest heart rate) was abnormal when ≤1.22 [6], or ≤1.17 in patients with permanent atrial fibrillation [14]. 

**Follow-up data.** The primary and only endpoint was all-cause mortality. Survival data were obtained from the departmental cardiology information system, which has regulated access to the governmental death registry database. Data at follow-up were complete for all patients. 

**Statistical analysis**. Standard statistical methods were used as previously detailed [6,7]. Outcome was assessed with Kaplan–Meier curves and Cox analysis, performed with Statistical Package for Social Sciences (IBM SPSS Statistics for Windows, Version 21.0. Armonk, NY, USA: IBM Corp.).

## 3. Results

Of 292 patients, 259 (89%) had angiographically smooth coronary arteries and 33 (11%) had coronary stenosis <50%. The main clinical, rest, and stress echocardiographic characteristics of the study patients are described in Table 1. 

**Stress echocardiography**. After dipyridamole, inducible RWMA were present in 20 (7%) patients. Heart rate significantly increased (from 70 ± 13 at rest to 90 ± 16 at peak of stress; *p* < 0.0001). HRR during dipyridamole SE was normal in 183 (63%) and abnormal in 109 patients (37%). Patients with impaired HRR showed higher heart rate at rest (*p* < 0.0001) and lower heart rate at peak stress (*p* < 0.0001) than patients with preserved HRR (Table 1). An example of an abnormal HRR in absence of inducible RWMA is reported in Figure 1.

**Outcome prediction.** During a follow-up of 10.4 ± 5.5 years, 89 patients (30%) died. 

Yearly mortality was similar in patients with angiographically smooth coronary arteries and those with coronary stenosis <50% (3.0 vs. 2.5%; *p* = 0.94).

The 15-year mortality rate was 27% in patients with normal and 54% in patients with abnormal HRR (*p* < 0.0001) (Figure 2). The univariate and multivariate prognostic indicators are shown in Table 2. In multivariable analysis, a blunted HRR during dipyridamole SE was an independent predictor of mortality (hazard ratio 1.86, 95% confidence intervals 1.20–2.88; *p* = 0.006) with age and permanent atrial fibrillation, while neither rest nor inducible RWMA were significant predictors. 

After exclusion of patients with permanent atrial fibrillation, age older than 70 years and under beta-blocker therapy, all of which were more prevalent in the reduced HRR group, patients with blunted HRR (*n* = 33) still showed a higher yearly mortality compared to those (*n* = 87) with normal HRR (2.5 vs. 0.8%; *p* = 0.01).

## 4. Discussion 

In patients with INOCA referred to testing, a reduced HRR is associated with worse survival, possibly due to a blunted sympathetic reserve elicited by endogenous adenosine accumulation determining an increased sympathetic drive [15,16].

### 4.1. Comparison with Previous Studies

HRR during dipyridamole SE targets the pathophysiological variable of cardiac autonomic function, which is important in determining electrical instability and arrhythmic vulnerability. Cardiac sympathetic dysfunction has been previously demonstrated to occur in as many as 30% of INOCA patients with different, more sophisticated electrocardiographic and cardiac imaging techniques [17,18,19,20,21,22]. Abnormal cardiac adrenergic nerve function also predicts unfavorable outcome, although this was assessed on a small population of only 40 patients and based on worsening of symptomatic status [23]. 

We previously reported that patients with INOCA are a heterogeneous population, and the minority developing stress-induced RWMA and reduced CFVR are at relatively higher risk [24,25]. In comparison with previous studies, the recruitment window in our study was from 1998 to 2019, while previous studies recruited different patients in a different time window, from 1983 to 2002 [24] or from 2002 to 2007 when coronary flow velocity reserve was first introduced in daily practice by early adopters [25]. This may account for the lower prognostic value of inducible RWMA in the present INOCA population. Over the last 40 years, we observed a decline in prevalence of inducible ischemia and an attenuation of the prognostic power of positive tests, possibly due to the increased number of tests performed under therapy and the widespread use of cholesterol-lowering, antihypertensive and antiplatelet drugs positively affecting prognosis in patients identified at higher risk after stress testing [26].

### 4.2. Clinical Implications

Chest pain in INOCA patients is a syndrome rather than a disease, and it recognizes very different mechanisms that must be identified to plan an effective therapy [27]. Dipyridamole SE identifies different phenotypes that are also, in theory, separate therapeutic targets focused on different mechanisms of disease. Stress-induced RWMA are a marker of angiographically occult coronary artery disease or initial cardiomyopathy, and can be targeted by anti-ischemic therapy. A reduced coronary flow velocity reserve is a hallmark of coronary microvascular dysfunction and is considered an indication of the use of drugs effective in protecting coronary microcirculation, such as angiotensin-converting enzyme inhibitors and statins. A third mechanism is identified by an abnormal HRR which is a biomarker of cardiac sympathetic dysfunction, largely unrelated to inducible ischemia or abnormal coronary microvascular function. The target cell is a cardiac afferent neuron, not the endothelium or the smooth muscle cell of coronary vessels. Sympathetic activation accelerates the heart rate via circulating adrenaline or neural release of noradrenaline [28], and the latter mechanism is more likely after dipyridamole. The blunted HRR is a risk biomarker, a specific phenotype, and a potential target of old and new drugs [29].

### 4.3. Study Limitations

This retrospective study was possible since the information on heart rate was stored in the data bank at the time of data acquisition and could be reanalyzed on the basis of new evidence linking HRR to cardiac sympathetic reserve and outcome.

All-cause death was the outcome measure, which is more objective and relevant than other soft endpoints [30]. We did not address the cause of death, which is more difficult to assess than simple all-cause death. We also did not include myocardial infarction among endpoints. However, it has been shown in the past that myocardial infarction can frequently occur in the absence of pre-existent significant coronary artery stenosis, and that the site of infarction is predicted by the site of inducible ischemia during SE only in a minority of cases [31]. 

At peak stress, SBP dropped on average of 8 mmHg in the abnormal HRR group and only 1 mmHg in the normal HRR group (*p* = 0.001). This finding might have attenuated the observed difference in HRR, since one would expect a compensatory rise in heart rate just from a drop in blood pressure. Beta blockers might have contributed to the blunted HRR, at least in theory, but it has been previously shown that resting heart rate, not HRR, are affected by concomitant beta-blocker therapy with vasodilator testing [7].

A minority of patients developed RWMA during stress. We did not perform a repeat coronary angiography in these patients, but it is well-established that some of these patients have angiographically occult forms of coronary artery disease that can be unmasked by techniques such as intravascular ultrasound or optimal coherence tomography study, allowing the assessment of the coronary lumen and wall more directly than the simple luminogram of coronary angiography. These patients also show a worse prognosis [32].

We did not include other parameters now part of the comprehensive stress echo protocol, such as coronary flow-velocity reserve or B-lines [13].

## 5. Conclusions

Dipyridamole SE is usually applied in INOCA patients to assess inducible ischemia with RWMA and abnormalities of coronary microcirculation. Imaging information can be expanded with imaging-independent assessment of HRR, which provides an index of cardiac autonomic balance and predicts survival better than inducible RWMA (Figure 3). HRR also represents a potential therapeutic target for pharmacological and nonpharmacological modulation of cardiac sympathetic tone. 

## Figures and Tables

**Figure 1 jcm-11-00052-f001:**
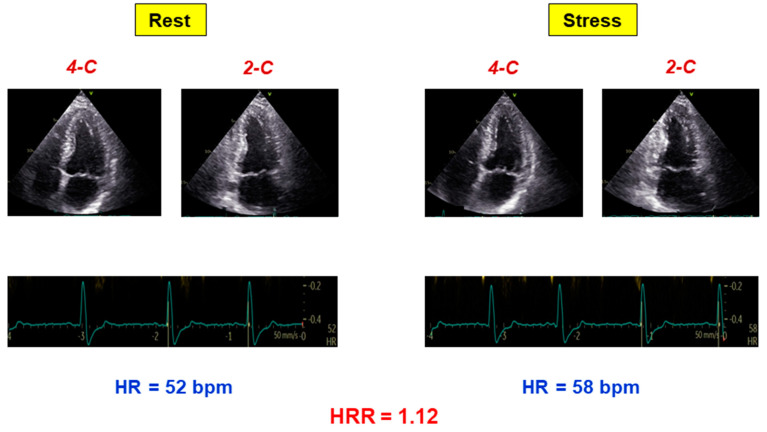
**Example of abnormal HRR.** A patient with no inducible RWMA and abnormal HRR (rest heart rate 52 bpm, stress heart rate 58 bpm; HRR 1.12). HR = heart rate; HRR = heart rate reserve.

**Figure 2 jcm-11-00052-f002:**
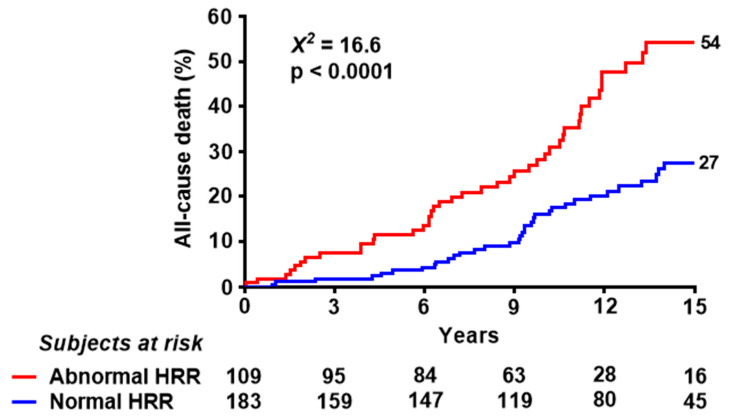
**Kaplan–Meier survival curves**. Kaplan–Meier survival curves (including all-cause death) in patients with normal and abnormal HRR. Number of patients per year is shown.

**Figure 3 jcm-11-00052-f003:**
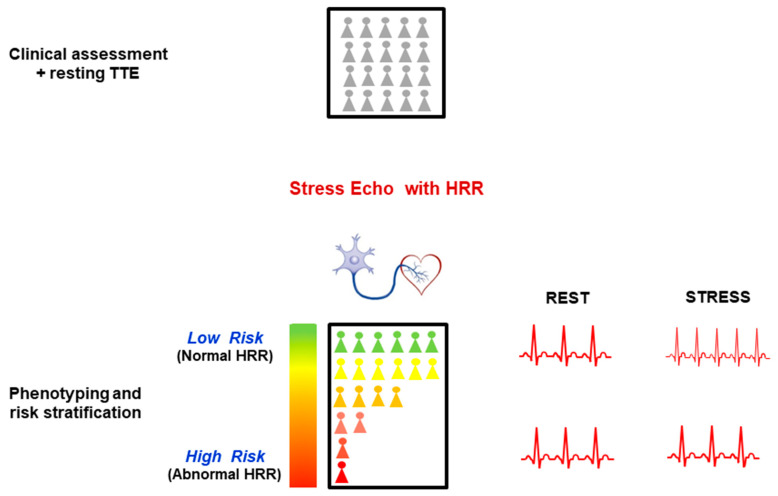
A population of allcomers with INOCA (*top panel*) is evaluated with dipyridamole SE, usually focused on myocardial ischemia (with stress-induced RWMA) and small vessels (with abnormal coronary flow-velocity reserve). The simultaneous assessment of HRR allows evaluation of cardiac autonomic unbalance with a simple one-lead ECG evaluating rest and stress heart rate (*lower right panels*). A blunted HRR identifies the sympathetic dysfunction phenotype, a higher risk subset and a potential therapeutic target.

**Table 1 jcm-11-00052-t001:** Clinical, echocardiographic, and prognostic findings of the study population.

	Abnormal HRR (*n* = 109)	Normal HRR (*n* = 183)	*p* Value
Age (years)	68 ± 10	62 ± 11	<0.0001
Males	63 (58%)	98 (54%)	0.48
** *Clinical history* **			
Diabetes mellitus	32 (29%)	39 (21%)	0.12
Arterial hypertension	74 (68%)	119 (65%)	0.62
Hypercholesterolemia	57 (52%)	96 (52%)	0.98
Cigarette smoking	31 (28%)	52 (28%)	0.99
Left bundle branch block	16 (15%)	14 (8%)	0.06
Permanent atrial fibrillation	12 (11%)	14 (8%)	0.33
Ongoing β-blocker therapy	46 (42%)	51 (28%)	0.01
** *Echocardiographic findings* **			
Rest ejection fraction (%)	56 ± 8	58 ± 7	0.03
Rest WMSI	1.18 ± 0.35	1.10 ± 0.26	0.02
Stress echo-induced RWMA	7 (6%)	13 (7%)	0.82
Rest HR (beats/min)	75 ± 14	67 ± 11	<0.0001
Peak HR (beats/min)	83 ± 15	94 ± 15	<0.0001
HRR	1.11 ± 0.10	1.42 ± 0.15	<0.0001
Rest SBP (mmHg)	138 ± 24	135 ± 19	0.18
Rest DBP (mmHg)	78 ± 13	77 ± 13	0.34
Peak SBP (mmHg)	130 ± 27	134 ± 21	0.15
Peak DBP (mmHg)	71 ± 14	74 ± 12	0.03
** *Follow-up data* **			
Duration of follow-up (years)	9.6 ± 5.3	10.8 ± 5.6	0.07
Deaths	47 (43%)	42 (23%)	<0.0001

Data presented are mean value ± SD or number (%) of patients. HRR = heart rate reserve; WMSI = wall motion score index; RWMA = regional wall motion abnormality; HR = heart rate; SBP = systolic blood pressure; DBP = diastolic blood pressure.

**Table 2 jcm-11-00052-t002:** Univariate and multivariate predictors of mortality.

	Univariate Analysis		Multivariate Analysis	
	HR (95% CI)	*p* Value	HR (95% CI)	*p* Value
Age (years)	1.09 (1.06–1.12)	<0.0001	1.08 (1.05–1.10)	<0.0001
Gender (male)	0.92 (0.61–1.40)	0.70		
Diabetes mellitus	1.40 (0.87–2.25)	0.16		
Arterial hypertension	0.90 (0.58–1.40)	0.65		
Cigarette smoking	0.81 (0.51–1.28)	0.36		
Left bundle branch block	1.27 (0.67–2.39)	0.46		
Permanent atrial fibrillation	3.36 (1.97–5.72)	<0.0001	2.70 (1.55–4.70)	<0.0001
Ongoing β-blocker therapy	1.25 (0.81–1.94)	0.31		
Rest ejection fraction	1.00 (0.97–1.03)	0.82		
Rest WMSI	0.96 (0.47–1.97)	0.91		
Stress echo-induced RWMA	1.03 (0.51–2.06)	0.94		
Abnormal HRR	2.34 (1.54–3.56)	<0.0001	1.86 (1.20–2.88)	0.006

HR = hazard ratio; CI = confidence interval. Other abbreviations as in Table 1.

## Data Availability

The data presented in this study are available on request from the corresponding author. The data are not publicly due to privacy.

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
