# Peer review of "Reduced Sympathetic Reserve Detectable by Heart Rate Response after Dipyridamole in Anginal Patients with Normal Coronary Arteries"

_jcm, 2021, doi:10.3390/jcm11010052_

Round 1

Reviewer 1 Report

Thank you for inviting me to review the paper titled "Reduced Sympathetic Reserve detectable by Heart Rate response after Dipyridamole in Anginal Patients with Normal
Coronary Arteries" by Cortigiani et al. The authors bring up an excellent viewpoint on dipyridamole stress acquired HRR as a surrogate marker for mortality in INOCA patients. The manuscript is written in simple language and answers the research question effectively. 

Author Response

Thank you for your encouraging comments.

Reviewer 2 Report

The authors present an interesting study on heart rate reserve in patients with evidence of ischemia and angiographically normal coronary arteries, and its correlation with all cause mortality. 

Regarding, the definition of the population angiographically normal was <50% stenosis. How many patients had 0% epicardial coronary disease? <50% does not mean "normal," but rather nonobstructive epicardial disease. Even some epicardial coronary disease will likely affect vascular and subsequent neurohormonal responses to external stimuli.

Regarding baseline demographics, there is a significant difference in the abnormal HRR group in age, use of beta blocker therapy and trend for atrial fibrillation. These are the same factors that give significant multivariate differences. All of the these significant factors interfere with each other. If age and atrial fibrillation are excluded, does the significance for HRR still hold?

In Table 1, there is an 8 point drop in SBP between rest and peak in the abnormal HRR group, but only 1 point drop in average SBP in normal HRR group. One would expect a compensatory rise in HR just from drop in BP. Do you think this is because of the beat blocker therapy?

In those patients with regional WMA/inducible ischemia, did any have repeat angiography? If so, was there disease that may have been underestimated on original angiogram, perhaps improved with additional IVUS/OCT modalities?

What was the cause of death? How many had myocardial infarction? And if there was myocardial infarction, how many nonobstructive CAD (0-50%) and how many had 0% disease?

Author Response

The authors present an interesting study on heart rate reserve in patients with evidence of ischemia and angiographically normal coronary arteries, and its correlation with all cause mortality. 

Regarding, the definition of the population angiographically normal was <50% stenosis. How many patients had 0% epicardial coronary disease? <50% does not mean "normal," but rather nonobstructive epicardial disease. Even some epicardial coronary disease will likely affect vascular and subsequent neurohormonal responses to external stimuli.

You are right. We added two sentences in results as follows:

Of 292 patients, 259 (89%) had angiographically smooth coronary arteries and 33 (11%) had coronary stenosis <50%.                                                                                                                

Yearly mortality was similar in patients with angiographically smooth coronary arteries and those with coronary stenosis <50% (3.0 vs 2.5%; p=0.94).

Regarding baseline demographics, there is a significant difference in the abnormal HRR group in age, use of beta blocker therapy and trend for atrial fibrillation. These are the same factors that give significant multivariate differences. All of the these significant factors interfere with each other. If age and atrial fibrillation are excluded, does the significance for HRR still hold?

This is a very important point that we addressed in results as follows:

Results. After exclusion of patients with permanent atrial fibrillation, older than 70 years and under beta-blocker therapy, all of which were more prevalent in the reduced HRR group, patients with blunted HRR (n=33) still showed a higher yearly mortality as compared to those (n=87) with normal HRR (2.5 vs 0.8%; p=0.01).

In Table 1, there is an 8 point drop in SBP between rest and peak in the abnormal HRR group, but only 1 point drop in average SBP in normal HRR group. One would expect a compensatory rise in HR just from drop in BP. Do you think this is because of the beat blocker therapy?

This is an important point that we addressed in the discussion as follows:

At peak stress, SBP dropped on average of 8 mmHg in abnormal HRR group and only 1 mmHg in the normal HRR group (p=0.001). This finding might have attenuated the observed difference in HRR, since one would expect a compensatory rise in heart rate just from a drop in blood pressure. Beta-blockers might have contributed to the blunted HRR, at least in theory, but it has been previously shown that resting heart rate, not HRR, are affected by concomitant beta-blocker therapy with vasodilator testing (7).

Discussion:

In those patients with regional WMA/inducible ischemia, did any have repeat angiography? If so, was there disease that may have been underestimated on original angiogram, perhaps improved with additional IVUS/OCT modalities?

We addressed this important point in study limitations as follows:

Discussion, study limitations. A minority of patients developed RWMA during stress. We did not perform a repeat coronary angiography in these patients, but it is well established that some of these patients have angiographically occult forms of coronary artery disease that can be unmasked by techniques such as intravascular ultrasound or optimal coherence tomography study allowing to assess coronary lumen and wall more directly than the simple luminogram of coronary angiography. These patients also show a worse prognosis (32).

  1. From AM, Kane G, Bruce C, Pellikka PA, Scott C, McCully RB. Characteristics and outcomes of patients with abnormal stress echocardiograms and angiographically mild coronary artery disease (<50% stenoses) or normal coronary arteries. J Am Soc Echocardiogr 2010;23:207-14.

What was the cause of death? How many had myocardial infarction? And if there was myocardial infarction, how many nonobstructive CAD (0-50%) and how many had 0% disease?

We addressed this important point in study limitations as follows:

Discussion, study limitations.We did not address the cause of death, which is more difficult to assess than simple all-cause death. We also did not include myocardial infarction among end-points. However, it has been shown in the past that myocardial infarction can frequently occur in the absence of pre-existent significant coronary artery stenosis, and that the site of infarction is predicted by the site of inducible ischemia during SE only in a minority of cases (31).

  1. Varga A, Picano E, Cortigiani L, Petix N, Margaria F, Magaia O, Heyman J, Bigi R, Mathias W Jr, Gigli G, Landi P, Raciti M, Pingitore A, Sicari R. Does stress echocardiography predict the site of future myocardial infarction? A large-scale multicenter study. EPIC (Echo Persantine International Cooperative) and EDIC (Echo Dobutamine International Cooperative) study groups. J Am Coll Cardiol 1996;28:45-51.

Round 2

Reviewer 2 Report

Thank you for addressing concerns from primary review. 

Author Response

Thank you for your encouraging comments.

This manuscript is a resubmission of an earlier submission. The following is a list of the peer review reports and author responses from that submission.